# Molecular Mechanisms and Key Processes in Interstitial, Hemorrhagic and Radiation Cystitis

**DOI:** 10.3390/biology11070972

**Published:** 2022-06-28

**Authors:** Clément Brossard, Anne-Charlotte Lefranc, Anne-Laure Pouliet, Jean-Marc Simon, Marc Benderitter, Fabien Milliat, Alain Chapel

**Affiliations:** 1Laboratoire de Radiobiologie des Expositions Médicales (LRMED), Service de Recherche en Radiobiologie et en Médecine Régénérative (SERAMED), Institut de Radioprotection et de Sûreté Nucléaire (IRSN), 92260 Fontenay-aux-Roses, France; clement.brossard@irsn.fr (C.B.); charlotte1810.lefranc@gmail.com (A.-C.L.); anne-laure.pouliet@irsn.fr (A.-L.P.); marc.benderitter@irsn.fr (M.B.); fabien.milliat@irsn.fr (F.M.); 2Radiotherapy Oncology Department, Hôpital Universitaire Pitié-Salpêtrière, 47-83 Boulevard de l’Hôpital, CEDEX 13, 75651 Paris, France; jean-marc.simon@aphp.fr

**Keywords:** cystitis, cancer, radiation, inflammation, bleeding, pelvic radiation disease, mechanism

## Abstract

**Simple Summary:**

Pathologies of the bladder are called cystitis. They cause discomfort for the patient. Due to persistent pain, bleeding, urinary incontinence, and uncontrolled urination, the chronic forms cause considerable degradation to patient quality of life. Currently, there is no curative treatment for the most severe forms. This is both an economic and a societal problem. Although the different forms of cystitis have different causes, they share common mechanisms. We propose to describe in detail the key processes and the associated mechanisms involved in abacterial cystitis.

**Abstract:**

Cystitis is a bladder disease with a high rate of prevalence in the world population. This report focuses on Interstitial Cystitis (IC), Hemorrhagic Cystitis (HC) and Chronic Radiation Cystitis. These pathologies have different etiologies, but they share common symptoms, for instance, pain, bleeding, and a contracted bladder. Overall, treatments are quite similar for abacterial cystitis, and include bladder epithelium protective or anti-inflammatory agents, alleviating pain and reducing bleeding. This review summarizes the mechanisms that the pathologies have in common, for instance, bladder dysfunction and inflammation. Conversely, some mechanisms have been described as present in only one pathology, such as neural regulation. Based on these specificities, we propose identifying a mechanism that could be common to all the above-mentioned pathologies.

## 1. Introduction

Cystitis is an acute or chronic inflammation of the bladder wall that may be related to infection, drugs, irradiation, or may be idiopathic. The most frequent cause of cystitis is infection, mainly by Escherichia coli [1]. In this article, we review the clinicopathologic features of Interstitial Cystitis, Hemorrhagic Cystitis and Radiation Cystitis, mainly comprising pelvic pain, incontinence, dysuria, and hematuria that may progress to hemorrhagic cystitis. Cystitis can be classified according to etiology and treatment approach into different subtypes such as traumatic, interstitial, eosinophilic, hemorrhagic, foreign body, emphysematous or radiation cystitis. In this review, we study the pathophysiological mechanisms of abacterial cystitis. We intend to define a common mechanism. Finally, we look ahead to a possible therapeutic approach to be investigated.

## 2. Structure and Function of the Bladder Wall

In order to fully appreciate the physiopathology of these diseases and the related mechanisms, an understanding of the structure of the urinary bladder wall is required.

The bladder wall is composed of the urothelium, the lamina propria, the detrusor, adventitia and serosa (Figure 1). The urothelium acts as a barrier against damaging solutes present in the urine and maintains osmotic pressure. The urothelium is made up of superficial cells, intermediate cells, basal cells, and basal stem cells. Superficial cells of the urothelium play a major role in the barrier function because these cells are in contact with the bladder lumen [1]. Basal cells contain stem cells able to regenerate the urothelium [2], which are basically quiescent until injury [3]. The lamina propria is located below the urothelium. The lamina propria may be subdivided into two parts, the upper lamina propria (ULP), just below the urothelium, and the deep lamina propria (DLP), between the ULP and the detrusor (Figure 1). The detrusor is composed of smooth muscles. This layer of muscles is divided into three, namely a longitudinal, a circular and another layer of longitudinal muscles allowing efficient bladder contraction. Adventitia and serosa form a very thin tissue layer, which covers the surface of the bladder.

## 3. Discussion

### 3.1. Etiology, Prevalence and Clinical Features of Abacterial Cystitis

#### 3.1.1. Interstitial Cystitis

Interstitial Cystitis (IC) is an idiopathic disease with chronic inflammation affecting the bladder (Table 1).

The disease is most likely multifactorial in origin and includes genetic parameters and environmental aspects. The prevalence estimates in the past decade now exceed 10 million US citizens (3–7% in women and 2–4% in men) [4,5].

Clinical characteristics are urinary frequency, urgency, and pain. Latency from symptom onset to diagnosis of IC is variable and can range from 1 month to 30 years, with an average of about 5 years [6,7,8,9,10,11]. Two forms of Interstitial Cystitis are described, the ulcerative form (called Hunner’s ulcer) and the non-ulcerative form. The ulcerative form accounts for 4 to 10% of IC cases. In the remainder of this review, we focus only on the non-ulcerative form of IC.

#### 3.1.2. Hemorrhagic Cystitis

Hemorrhagic Cystitis (HC) is defined as a diffuse inflammatory disease of the bladder resulting in hemorrhage of the bladder mucosa.

Irradiation and chemotherapy using cyclophosphamide (CP) as treatment are the most common etiological agents [11,12,13,14].

The frequency of patients developing HC is 5–10% in the 6–20 years after radiation therapy [12] and 20–25% in the weeks or months after CP chemotherapy [13].

HC can progress to a severe hemorrhagic syndrome, resulting in loss of bladder function. To compare IC and CRC to HC, in the remainder of the review, we focus on cyclophosphamide-induced HC.

#### 3.1.3. Chronic Radiation Cystitis

Chronic Radiation Cystitis (CRC) is a lesion of the bladder resulting from irradiation of a pelvic organ, particularly following radiation therapy (prostate, colon or cervix).

The chronic phase of Radiation Cystitis (CRC) may appear from 6 months to 20 years in 5–10% of patients after radiation therapy [13,14,15,16,17,18,19,20].

CRC is characterized by microscopic or macroscopic hematuria accompanied by pain and pollakiuria. Lesions range from a simple inflammatory reaction to almost complete loss of bladder function. The molecular mechanisms of CRC are insufficiently described in the literature, particularly regarding inflammatory processes and fibrosis.

## 4. Molecular Mechanisms in Cystitis

The molecular mechanisms of the different forms of cystitis are quite similar except for the neural aspect. Chronologically, they include urothelium dysfunction, inflammation, and vascular dysfunction followed by fibrosis. These mechanisms occur in cascades, and they self-amplify. The consequence is an aggravation of these pathologies over time. In the section below, we have classified our review according to these events and examined, first, the common then the specific mechanisms.

### 4.1. Bladder Mucosa Dysfunction

#### 4.1.1. Common Mechanism

The loss of permeability of the bladder mucosa is the initiating point of cystitis. Although the agents of aggression are different, the consequence is a degradation on the entire bladder wall, which generates the other processes, namely inflammation, vascular attack and finally fibrosis. Below, we describe for each pathology the specific mechanisms responsible for the loss of impermeability of the bladder mucosa (Table 2).

#### 4.1.2. Specific Mechanisms

##### Interstitial Cystitis

During Interstitial Cystitis, the bladder mucosa undergoes modifications such as thinning, as well as partial degradation of the GAG layer (Figure 2). These changes alter permeability, allowing urinary solutes such as potassium to infiltrate the bladder wall [21].

An increase in adenosine triphosphate (ATP) concentration or a decrease in nitric oxide (NO) may impair the bladder contraction function [22]. These alterations lead to an increase in antiproliferative factor (APF) expression and cell apoptosis [23]. APF inhibits cell proliferation and prevents repair of damaged bladder mucosa [24], contributing to increased permeability. APF may also decrease the formation of tight junctions in urothelial cells [25]. Patients have a reduced expression of the tight junction proteins like zona occludens 1 and E-cadherin [26,27,28,29,30,31].

##### Hemorrhagic Cystitis

During Hemorrhagic Cystitis, dysfunction of the bladder mucosa is observed, however, the mechanism is different from that seen in IC (Figure 3). During chemotherapy, a portion of CP or ifosfamide is metabolized in the liver to acrolein, which is then excreted by the kidneys and stored in the bladder until urination [32]. This metabolite is known to be the causative agent of CP-induced HC [33]. Acrolein causes damage manifested by subepithelial edema, neutrophil infiltration, cell death, neovascularization, hemorrhage and ulceration. These mechanisms are described in Figure 3.

##### Chronic Radiation Cystitis

In the first phase of Chronic Radiation Cystitis, a decrease in the number of superficial cells is observed [34,35]. In the late phase, the bladder mucosa shows structural lesions (Figure 4). In different animal models, hyperplasia, erosion of the urothelium, atrophy and cellular edema (especially in the basal cells) have been described. A decrease in the expression of both uroplakin III and E-cadherin in superficial cells induces a decrease in bladder tightness. Hyperplasia of intermediate cells remains the most frequently observed phenomenon [34,35,36,37]. Infiltration of urine into the tissues maintains inflammation and urothelium damage [34,35], as described in Figure 4.

### 4.2. Inflammation

#### 4.2.1. Common Mechanism

In these different forms of cystitis, mastocytes, telocytes and fibroblasts are the key effectors, with a common pathway initiated by cytokines produced by damaged urothelium. However, the mechanisms seem to diverge, and their specificity is described for each form of cystitis below (see Table 2).

#### 4.2.2. Specific Mechanisms

##### Interstitial Cystitis

In Interstitial Cystitis, damaged urothelium initiates production of the following inflammation cytokines and chemokines: stem cell factor (SCF), interleukin-8 (IL-8), interleukin-1β (IL-1β), interleukin-6 (IL-6), tumor necrosis factor α (TNF-α), nerve growth factor (NGF) and chemokine ligand 2 (CCL2) [23,38,39]. Telocytes are active players and intervene early in inflammation. Telocytes and myofibroblasts produce cytokines and other molecules that may recruit immune cells [40,41,42]. Infiltration of urine mediates depolarization of nerve and muscle cells and activation of mast cells [43]. During this process, the release of neurotransmitters by the urothelium, such as adenosine triphosphate (ATP), acetylcholine, and nitric oxide (NO), induces defective communication between the mucosa, the muscle layer and the afferent neuronal system. SCF stimulates mast cell proliferation and activation, particularly in ulcerous IC [44]. This release of mediators may explain the painful symptoms of IC [45,46]. Tryptase may induce microvascular leakage and reactivate mast cells [47], leading to further urothelial damage, chronic inflammation, and pain [48]. Inflammation induces edema, generating an increase in volume in the lamina propria, particularly in ULP [49,50]. The thickening of ULP is heterogeneous, principally in the lower ULP [50,51,52] and foci of hypersensitivity may alternate with foci of hyposensitivity, compromising the correct integration of related stimuli [53]. Chronic inflammation is associated with elevated serum CRP levels and decreased urinary tract symptoms [54].

##### Hemorrhagic Cystitis

In Hemorrhagic Cystitis, initially, acrolein causes an increase in pro-inflammatory mediators [55], and cyclooxygenase-2 (COX-2) and reactive nitrogen species (RNS) into the urothelium [56]. Acrolein chronologically causes ROS production in the bladder epithelium [57] and induces inducible nitric oxide synthases (iNOS), resulting in NO overproduction. NO toxicity is thought to result from the formation of RNS [58].

The consequences of overproduction of ROS and RNS during inflammation are oxidative stress, cell damage and cell death (apoptosis/necrosis) [59]. Nuclear transcription factor-kappa B (NF-κB) and activator protein 1 (AP-1) are then induced [57]. Their activation induces expression of pro-inflammatory cytokines (TNF-α, IL-1β), induction of iNOS and increased production of ROS. These cytokines cause neutrophil infiltration at the inflammatory site, which may potentiate the inflammation [60].

##### Chronic Radiation Cystitis

In Chronic Radiation Cystitis, radiation-induced damage to epithelial cells releases pro-inflammatory factors that activate mast cells [61]. These cells actively participate in the production of pro-inflammatory cytokines and in vascular lesions. The presence of mast cells at the end of the early phase [37], and their persistence during the late phase [35], may favor chronic inflammation.

### 4.3. Vascular Response

#### 4.3.1. Common Mechanism

In the forms of cystitis investigated, vascular response is a central factor common to all three. VEGF causes vasodilation and immature angiogenesis, resulting in microhemorrhagic vessels and hypoxia. Secondly, telangiectasia and albumin promote chemokine production. This production mediates the activation of mast cells which induces neo-vascularization and an increased permeability of blood vessels [62]. The consequence is the localization of albumin outside the vessels, promoting the recruitment and activation of new mast cells. It is therefore a mechanism that involves both inflammation and angiogenesis to cause the leakage phenomenon. However, studies on the vascular response in specific pathologies have revealed complementary data, which are outlined below.

#### 4.3.2. Specific Mechanisms

##### Interstitial Cystitis

In Interstitial Cystitis, a high concentration of VEGF in the bladder induces vasodilatation and immature angiogenesis, which may be responsible for the hypervascularization and glomerulations characteristic of IC [63,64]. Glomerulations are associated with overexpression of hypoxia-inducible factor 1 α (HIF-1α) [65]. IL-16, IL-18, stem cell growth factor β (SCGFβ), cutaneous T-cell-attracting chemokine (CTACK), tumor-necrosis-factor related apoptosis inducing ligand (TRAIL), intercellular adhesion molecule 1 (ICAM-1), monocyte-chemotactic protein 3 (MCP-3) and vascular cell adhesion molecule 1 (VCAM-1) expression significantly increase in the bladder wall of IC patients [66].

##### Hemorrhagic Cystitis

In Hemorrhagic Cystitis, a high concentration of VEGF has been found in the bladder of CP-treated rats along with a significant positive regulation of ICAM-1 gene expression [67]. Moreover, increased levels of MCP-1 and VCAM have been observed in a model of acute cystitis induced by CP in rats [68].

##### Chronic Radiation Cystitis

In Chronic Radiation Cystitis specifically, a significant decrease in vascular density in the cranial part of the bladder (the dome) has been reported [69]. In preclinical models, a decreased number of blood vessels without a decrease of surface has been observed [16,17]. This can be explained by the permanent increase in microvasculature diameter with thickening of the vessel wall (telangiectasia) present in the tissues [8,53]. This telangiectasia has been highlighted [35,61,70,71], with increased permeability of the blood vessels and immuno-histological observation of albumin around the small vessels [70]. Vascular lesions, telangiectasias and the presence of albumin can result in the activation of macrophages and mast cells, resulting in the release of large quantities of tryptase, which amplifies vascular lesions and fibrosis [45,47]. These alterations, as they worsen, may induce microscopic and then macroscopic hematuria as observable in CRC (Table 3).

### 4.4. Fibrosis

#### 4.4.1. Common Mechanism

Regarding Interstitial, Hemorrhagic and Radiation cystitis, a common feature is fibrosis, which may result in decreased contractile bladder capacity and bladder rigidification. Contractions are more difficult, resulting in urination dysfunction. Persistent inflammation is a prevalent denominator of fibrosis in these forms of cystitis. Fibrosis is caused by positive regulation of collagen I and III, fibronectin and TGFβ1 genes, negative regulation of WNT11 and production of YKL-40 by mast cells in IC. Based on these similarities, we can hypothesize that YKL40 may also play a key role, mainly due to mast cells present in large numbers in all forms of the pathology. Fibrosis invades the detrusor and lamina propria, particularly around the blood vessels, making them more rigid. The consequence is a decrease in bladder functionality and difficulty with contractions, producing spasms and emergencies, etc. Telocytes participate in the regulation of contractions, which may contribute to incontinence and spasms. Spasms favor rupture of the vessels, leading to microscopic hematuria and then to macroscopic hematuria, fostering inflammation, promoting fibrosis and then vascular lesions. Below, we describe in detail the specificity of each pathology with regard to fibrosis (Table 3).

#### 4.4.2. Specific Mechanisms

##### Interstitial Cystitis

During Interstitial Cystitis, progressive pro-fibrotic changes in the bladder are mediated by the secretion of transforming growth factor β1 (TGF-β1) which induces the trans-differentiation of fibroblasts into myofibroblasts mediating an excessive deposition of extracellular matrix (ECM) (collagen I and III, fibronectin) in the lamina propria and smooth muscle [72]. These changes are associated with the positive regulation of collagen I and III genes [73], fibronectin [74], TGF-β1 [39,75], YKL-40 [76,77,78], and the negative regulation of sonic hedgehog (Shh), genes from the WNT family [74]. YKL-40 (also known as CHI3L1 for chitinase-3-like protein 1) is expressed in detrusor mast cell granules and submucosal macrophages. A higher expression of WNT2B and WNT5A has been quantified in patients with ulcerative IC. Conversely, negative regulation of WNT11 leads to pro-fibrotic changes in the bladder epithelial cells of patients with non-ulcerative IC [74]. Mast cells could amplify fibrosis [76,77,78]. Fibrosis and inflammation (with edema) contribute to this enlargement of the network’s mesh, which favors spasm and incontinence because of disrupted communication between the telocytes and nerve fibers. 

##### Hemorrhagic Cystitis

In chronic cases of Hemorrhagic Cystitis, fibrosis of the bladder wall may develop progressively [17].

##### Chronic Radiation Cystitis

Throughout Chronic Radiation Cystitis, TGF-β is strongly expressed in the urothelium and lamina propria (submucosa), and diffusely in the muscle layers [79]. An increase in the expression of collagen I and III is measured mainly in the submucosa and muscle [35,69,71,79,80]. Infiltration of the ECM between muscle fibers is observed, which is homogeneous in the three parts of the bladder (dome, body and trigone) [69,80]. The increase in collagen density and the invasion of the muscles by the ECM leads to a loss of muscle and contractile capacity of the bladder. The consequence is incomplete, abrupt and involuntary bladder contractions [80]. The loss of contractility results in increased urinary frequency but no change in urine volume. Urinary frequency is correlated with a decrease in total bladder volume [81,82,83,84].

### 4.5. Assumptions Regarding Central Mechanisms in Interstitial, Hemorrhagic and Radiation Cystitis 

The hypotheses used to describe the central mechanisms involved are divided into four phases (Figure 5). The first phase is a decrease in the impermeability of the urothelium due to erosion, hyperplasia, decreased uroplakin III, cellular edema or atrophy. In a second phase, the secretion of inflammatory factors produced by urothelial lesions (TNF-α, IL1-β, IL-6, IL-8 and CCL2) induce recruitment and activation of immune cells. Mast cells secrete VEGF and tryptases, inducing vascular damage. The consequences are imperfect vascular regeneration, leading to telangiectasia, increased vascular permeability, and hematuria, causing the release of albumin. This release amplifies the secretion of pro-inflammatory factor which increases inflammation and may lead to the maintenance of urothelial lesions. Mast cells could also act on the hyperactivation of nerve fibers through the secretion of NGF. A third phase might be the proliferation of nerve fibers, particularly those containing the neuropeptide substance P (SP). An increased reactivity to SP released by the perivascular sensory terminals could lead to a local cascade of neurogenic inflammatory responses that triggers pain. Edema and fibrosis enlarge this network, inducing miscommunication with the afferent nerves which could explain the spasms. This neurogenic inflammation reinforces chronic inflammation. T cells could also induce fibrosis through the secretion of TGF-β. The fourth phase entails myofibroblast formation and excessive deposition of ECM (collagen I and III, fibronectin) in the submucosa and smooth muscle. In addition, mast cells may amplify fibrosis through the production of YKL-40. This production of YKL-40 may increase the accumulation of ECM. Telocytes participate early in inflammation and may induce fibrosis by adopting a myofibroblast phenotype producing extra cellular matrix. Fibrosis leads to non-functional contractions that promote blood vessel damage, leading to further inflammation.

Treatments aim to reduce inflammation, vascular damage, fibrosis, urothelial damage and neuroinflammation (Figure 5). To reduce the loss of impermeability of the urothelium, glycosaminoglycan-based treatments have been used (mainly pentosan polysulphate). However, treatment times (over 6 months) and the high relapse rate (23% at 1 year) are significant limitations [85,86,87,88,89,90]. To reduce inflammation, drugs such as hydroxyzine, cimetidine and corticosteroids have been used. Hydroxyzine and cimetidine are used to reduce mast cell activation. These treatments have a low level of efficacy (<50%) and their effects stop when treatment is stopped [88,91,92]. As for corticoids, their action remains limited in time [93,94]. In order to reduce hemorrhages, intravesical instillation of formalin or alum can be used. The effect of this is the development of a vasoconstriction that reduces bleeding. These two techniques are less effective (<50% with recurrence rates of 10 to 25% over 6 months) and painful for the patient [85,88,95,96,97]. One last and more promising technique is hyperbaric oxygen therapy. This technique is designed to increase the supply of oxygen to stimulate angiogenesis. This has a success rate of 75% at 6 months, decreasing over time to 40% after a few years [91,98,99,100]. No anti-fibrotic drugs are recommended by the US Food and Drug Administration, but various preclinical treatments have been used, such as N-acetylcysteine and tipelukast [101,102].

To treat neuroinflammation, solutions such as dimethylsulfoxide (DMSO) or tricyclic antidepressant drugs have been used. DMSO has an analgesic effect on the afferent nerves. Despite a high level of efficacy (between 61 and 95%) [103], its limitations are an increase in urothelium lesions and increased fibrosis [85,87,88,91,103,104,105]. Tricyclic antidepressants (amitriptyline) have an efficacy rate of 50% to 66%, but decrease patients’ quality of life (causing drowsiness and nausea) [106]. Lastly, hydrodistension can be used. This consists of distending the bladder to break the nerve pathways before reconstructing them. However, low efficacy (56% of cases) and the risk of aggravation (9%) limit the use of this technique [91,107,108]. The majority of the drugs mentioned in this review only act on one component of the system. It is therefore essential to provide patients with new therapeutic solutions that will act simultaneously and permanently on the multiple different mechanisms (e.g., inflammation and loss of impermeability) of cystitis described in this review. Cell therapy based on mesenchymal stem cells (MSCs) could be an effective and inexpensive alternative. MSCs induce increased tissue regeneration (epithelium, endothelium), induce immunomodulation, and limit fibrosis. MSCs have been used in a clinical setting to treat HC and CRC and have shown a complete reduction in bleeding and pain in patients following a single injection [109,110,111]. In the case of IC, many clinical trials are underway. MSCs are very promising and further studies on the mechanisms involved may provide insight into their mode of action.

## Figures and Tables

**Figure 1 biology-11-00972-f001:**
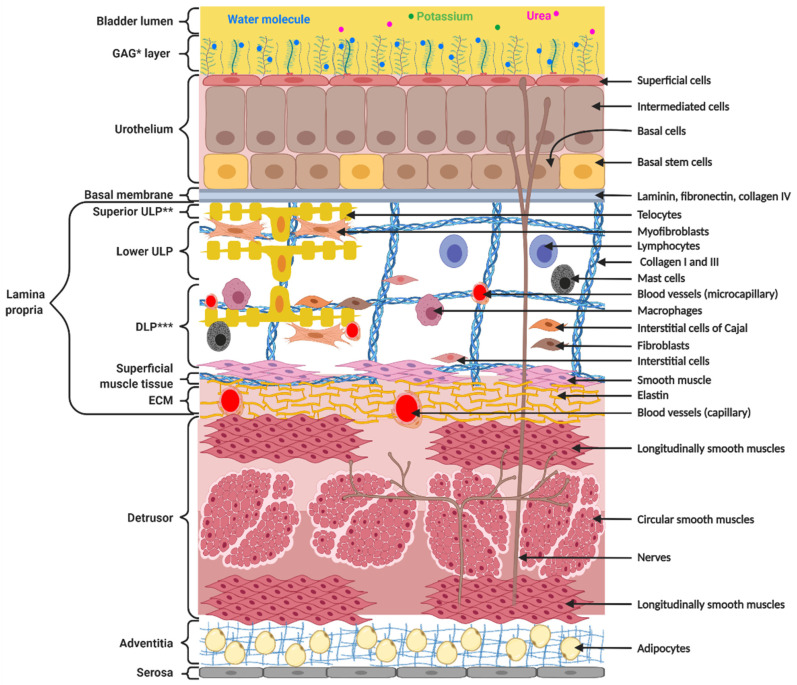
Structure and composition of the bladder wall. The bladder wall is composed of the urothelium, providing impermeability, the lamina propria, ensuring flexibility and compliance, the detrusor, allowing contractions, and the adventitia and serosa, protecting the bladder. * GAG = Glycoaminoglycan; ** ULP = upper lamina propria; *** DLP = deep lamina propria.

**Figure 2 biology-11-00972-f002:**
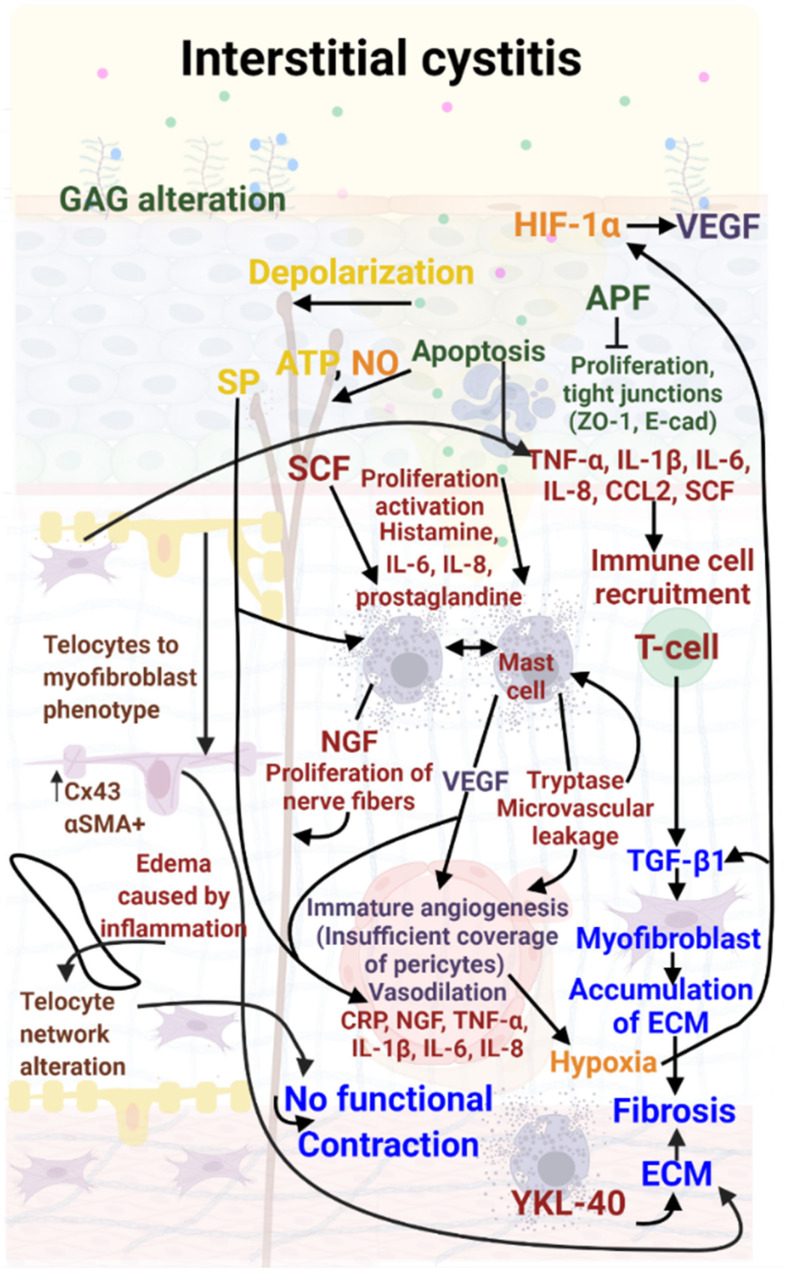
Molecular mechanisms involved in Interstitial Cystitis (IC). Bladder mucosa lesions induce inflammation mediated by IL-6, IL1 β, IL 8, CCL2 and TNF α to recruit T cells and mast cells. Mast cells produce tryptases and VEGF inducing vascular injury, creating hematuria and reinforcing inflammation. T lymphocytes produce TGF-β 1 inducing the transdifferentiation of fibroblasts into myofibroblasts. These myofibroblasts produce large amounts of ECM inducing fibrosis. Nerve damage due to potassium in urine and NGF produced by mast cells induces incomplete and intermittent contraction. Color code: red: inflammation; green: urothelial damage; blue: fibrosis, purple: vascular damage; orange: hypoxia, yellow: neuro-inflammation; brown: telocyte damage.

**Figure 3 biology-11-00972-f003:**
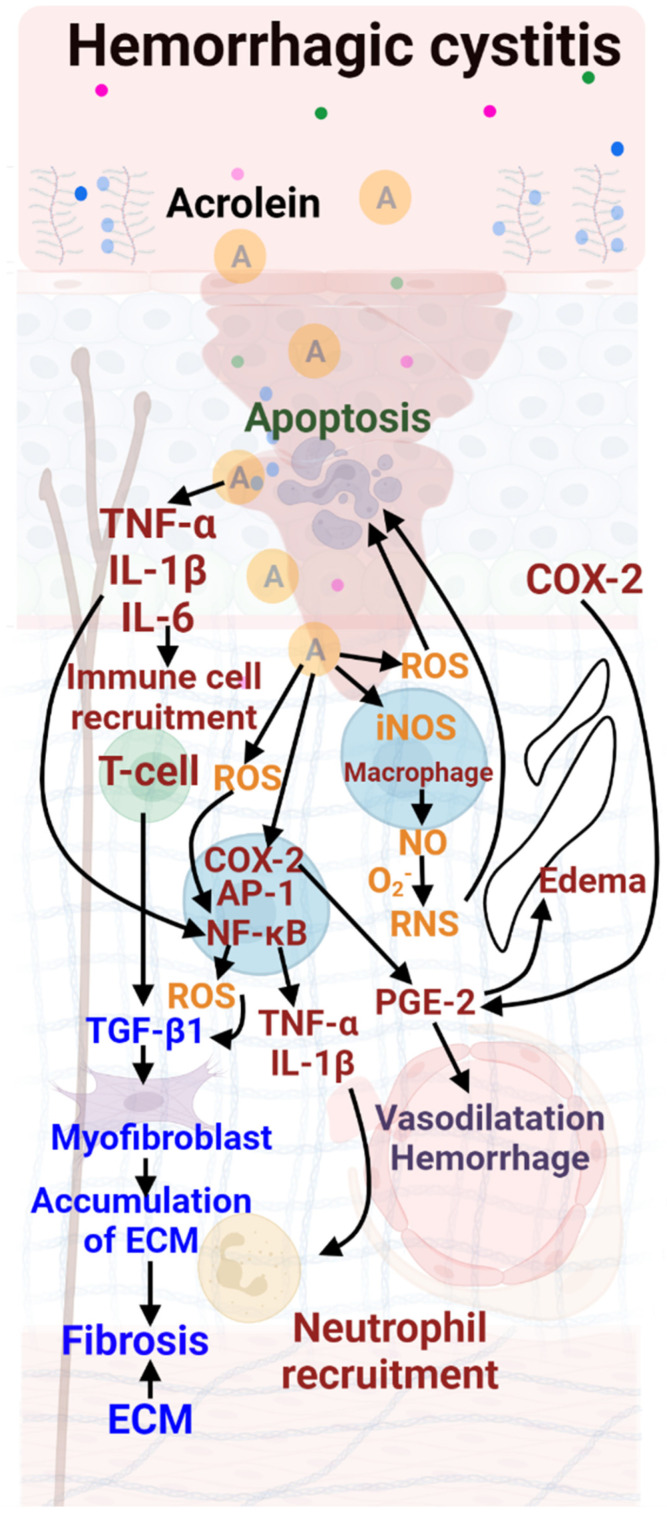
Molecular mechanisms involved in Hemorrhagic Cystitis (HC). Supplementary lesions are described as edema due to COX-2, which induces PGE-2. PGE-2 also induces vascular damage. A neutrophil recruitment is observed. Color code: red: inflammation; green: urothelial damage; blue: fibrosis; purple: vascular damage; orange: hypoxia.

**Figure 4 biology-11-00972-f004:**
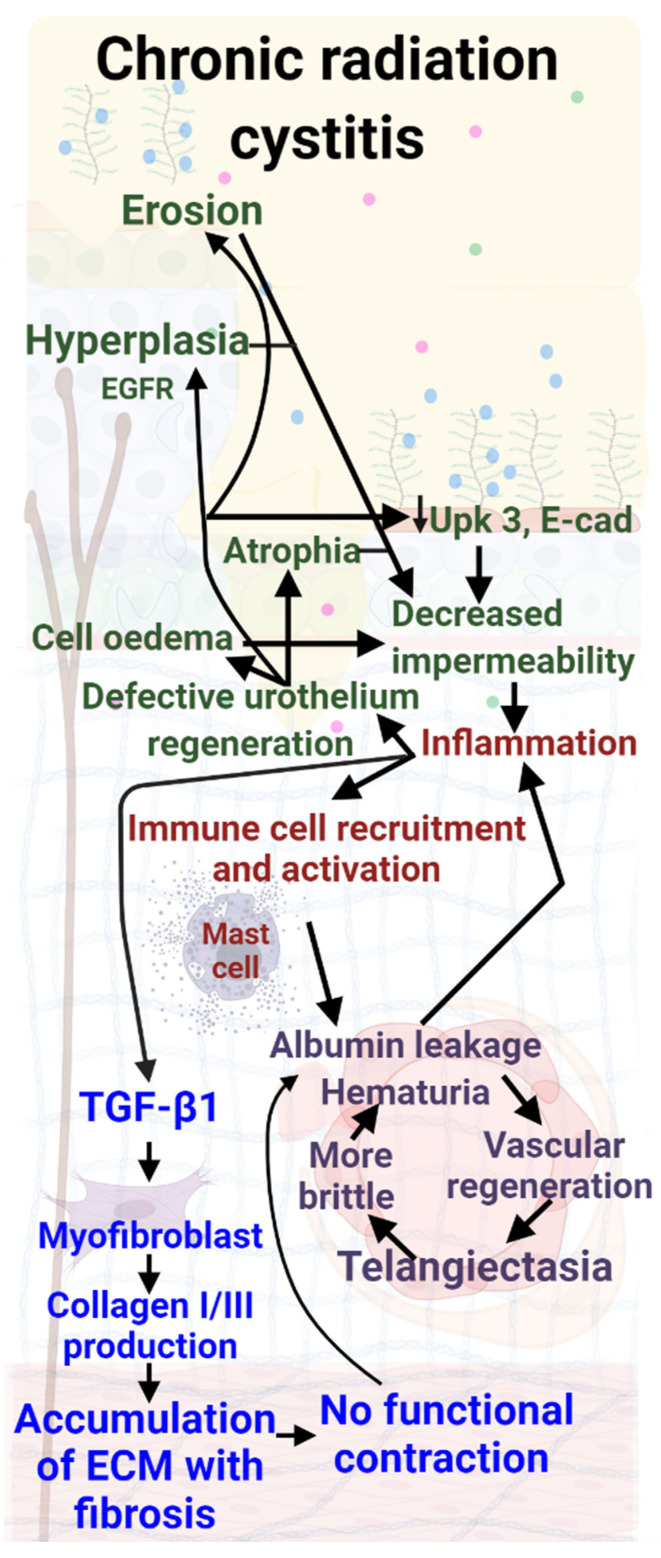
Molecular mechanisms involved in Chronic Radiation Cystitis (CRC). Several bladder mucosa lesions are described, such as hyperplasia, erosion, or atrophy. As far as inflammation is concerned, only the role of mast cells has been described. Fibrosis with collagen accumulation is defined but neither myofibroblast nor T-lymphocyte implication has been described; only an increase in TGF-β 1 production has been reported. Color code: red: inflammation; green: urothelial damage; blue: fibrosis; purple: vascular damage.

**Figure 5 biology-11-00972-f005:**
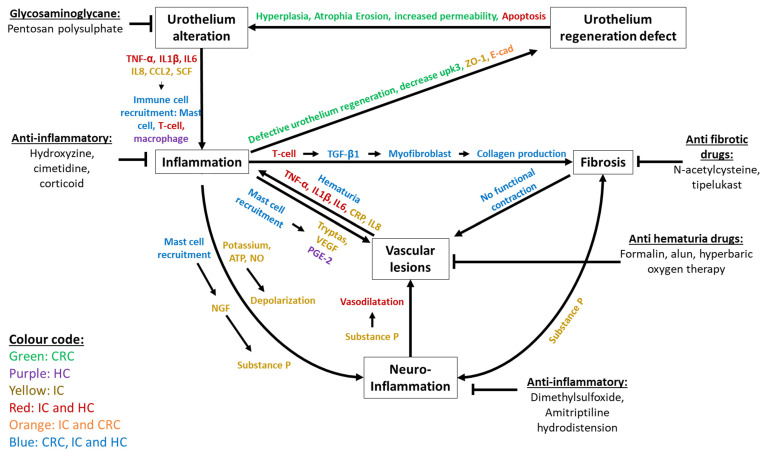
Synthetic description of the key processes and mechanisms shared by Interstitial, Hemorrhagic and Chronic Radiation Cystitis, including effectors and treatments. Color code: Green is used for effectors in CRC only, purple for those in HC only and yellow for those in IC only. Effectors found in IC and HC are shown in red and the effectors found in IC and CRC are shown in orange. The effectors common to CRC, IC and HC are shown in blue.

**Table 1 biology-11-00972-t001:** Main features of each type of cystitis.

Pathology	Interstitial Cystitis	Hemorrhagic Cystitis *(Not a Distinct Form of Cystitis)*	Chronic Radiation Cystitis
**Causes**	Idiopathic disease	Radiation, Chemotherapy (cyclophosphamide and ifosfamide), Infections (bacterial, viral, fungal and parasitic), Drugs, Toxins, Idiopathic diseases, Organ transplant	Pelvic radiation
**Signs**	Pain, Day and night pollakiuria, Urinary urgency		
		Dysuria, Incontinence, Hematuria (microscopic to severe macroscopic with clots)	
**Symptoms**	Dyspareunia Digestive and/or gynecological disorders Glomerulations	Severe form: urinary obstruction and acute renal failure	
**Diagnosis**	Cytobacteriological examination, Physical examination, Biopsy		
	Urodynamic examination Cystomanometry Voiding diary Questionnaires	Ultrasound	

**Table 2 biology-11-00972-t002:** Pathophysiological mechanisms: Urothelium dysfunction and inflammation.

Mechanism	Interstitial Cystitis	Hemorrhagic Cystitis * *(* Not a Distinct Form of Cystitis)*	Chronic Radiation Cystitis
**Urothelium dysfunction**	Degradation of the GAG layer Alteration of permeability causing urinary potassium infiltration leading to activation of mast cells and depolarization of nerve and muscle fibers Alteration of the ATP/NO ratio Abnormal expression of APF inhibiting cell proliferation and the formation of tight junctions and preventing urothelium repair Cellular apoptosis	Urothelium degradation, subepithelial edema and ulceration Alteration of permeability	Decrease in the expression of uroplakin III and E-cadherin leading to a decrease in impermeability Hyperplasia, atrophy and/or erosion of urothelium Cellular edema (mainly in basal cells)
**Inflammation**	Production of IL-1β, IL-6, TNF-α		
	Production of SCF, IL-8 and CCL2 by urothelium SCF-stimulated proliferation and activation of mast cells Degranulation of mast cells releasing histamine, IL-6, IL-8, prostaglandins, VEGF, NGF and tryptases Microvascular leakage and tryptase activation of mast cells Vasodilatation and immature angiogenesis caused by VEGF Insufficient pericyte coverage resulting in hemorrhagic vessels and hypoxia Over-expression of HIF-1α Increased expression of IL-16, IL-18, SCGFβ, CTACK, TRAIL, ICAM-1, MCP-3 and VCAM-1 in the bladder wall	Increase in COX-2 at the urothelium level Oxidative stress, cell damage and apoptosis/necrosis due to overproduction of ROS and RNS Induction of NF-κB and AP-1 in resident bladder cells due to the production of ROS, IL-1β and TNF-α Induction of iNOS leading to overproduction of NO and production of ROS caused by NF-κB and AP-1 Neutrophil Recruitment	Increased number of mastocytes in tissue No study on molecular mechanisms

**Table 3 biology-11-00972-t003:** Pathophysiological mechanisms: Neural regulation, vascular lesions and fibrosis.

Mechanisms	Interstitial Cystitis	Hemorrhagic Cystitis * *(* Not a Distinct Form of Cystitis)*	Chronic Radiation Cystitis
**Neural** **regulation**	Release of substance P (SP) from C-fibers leading to vasodilation and degranulation of mast cells Increased nerve fiber proliferation caused by NGF Overexpression of SP receptor mRNA	Unknown	Unknown
**Vascular** **lesions**	High concentration of VEGF Overexpression of ICAM-1, VCAM-1		Telangiectasias Albumin leakage Micro- and macro- hematuria
	Vasodilatation and immature angiogenesis Hypervascularization Glomerulations Overexpression of HIF-1α, IL-16, IL-18, SCGFβ,CTACK, TRAIL, MCP-3	Overexpression of MCP-1	
**Fibrosis**	Increased TGF-β1expression Excessive deposition of ECM in the lamina propria (submucosa) and smooth muscle Positive regulation of collagen I and III,		
	Myofibroblast formation		
	Positive regulation of fibronectin Negative regulation of WNT11 Production of YKL-40 causing ECM accumulation Decreased contractile bladder capacity and bladder stiffening		

## Data Availability

Not applicable.

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
