# Peer review of "Molecular Mechanisms and Key Processes in Interstitial, Hemorrhagic and Radiation Cystitis"

_biology, 2022, doi:10.3390/biology11070972_

Round 1

Reviewer 1 Report

The manuscript is interesting.

However I think some improvements are to be recommended:

1) FIgures should be more clear; I think the figures contain too many indications and they probably need to be enlarged to allow a better comprehension by the readers;

2) English should be reviewed and also some grammatical errors and typing errors need to be erased (in the Introduction the word “propose” needs the “S”; at row 90 please put the “T” before the “he”; change “miction” into “micturition”, and please review the English style possibly with the contribute of official language reviewers; 

3) I think that people reading manuscripts discussing molecular basis of cystitis already know the bladder anatomy; I therefore think that the macroscopic informations of bladder anatomy should be removed from the paper; please keep in the manuscript morphologic data (microscopic, ultrastructural) that help the reader to understand the pathophysiology of cystitis; 

4) I think the sub-division in too many pargraphs of the manuscript could risk to confound and to bore the reader. I would try to provide a better organization of the paper resuming into a paragraph (for example in a “discussion paragraph” the pathophysiologic phases of the condition and the related molecular events) and reduce the other paragraphs.

5) I suggest to the Authors to highlight what’s the contribute of this manuscript to the Literature. I’m not sure that it contains many additional informations to what is already described in specific articles already published. Therefore, if the goal of this paper ia to resume data regarding molecular traffic in the bladder inflammation I think it is very very important to achieve a more readable paper (better structured).

6) I would try to speculate future therapeutic targets for cystitis basing on the data reported in this paper. 

I hope my comments could help You to improve the quality of this manuscript.
Please follow my suggestions and those of the other reviewers to achieve a better outcome.

Author Response

We are grateful for the reviewer's help and accuracy. We have carefully considered the reviewer’s comments and advice and performed new data and revised our manuscript accordingly.

We believe that these changes have further strengthened the manuscript (highlighted in yellow).

  1. Figures should be more clear; I think the figures contain too many indications and they probably need to be enlarged to allow a better comprehension by the readers;

We thank you for your comments and apologize for figures. Both figures have been revised. Figure 2 has been modified to make it more understandable and divided into three independent figures in order to enlarge the figures and improve understanding. A colour code has been added to each figure.

  1. English should be reviewed and also some grammatical errors and typing errors need to be erased (in the Introduction the word “propose” needs the “S”; at row 90 please put the “T” before the “he”; change “miction” into “micturition”, and please review the English style possibly with the contribute of official language reviewers;

The English was revised by a company specialized in the correction of scientific articles (https://www.caupenne-co.com/en/).

  1. I think that people reading manuscripts discussing molecular basis of cystitis already know the bladder anatomy; I therefore think that the macroscopic informations of bladder anatomy should be removed from the paper; please keep in the manuscript morphologic data (microscopic, ultrastructural) that help the reader to understand the pathophysiology of cystitis;

The structure of the bladder has been removed from the text and from Figure 1.

  1. I think the sub-division in too many pargraphs of the manuscript could risk to confound and to bore the reader. I would try to provide a better organization of the paper resuming into a paragraph (for example in a “discussion paragraph” the pathophysiologic phases of the condition and the related molecular events) and reduce the other paragraphs.

We thank you for this suggestion that improves this article. Following your recommendations, we have restructured the article with a discussion paragraph and reduced the other paragraphs. A summary associated with a figure (figure 4) has been added to help understanding.

  1. I suggest to the Authors to highlight what’s the contribute of this manuscript to the Literature. I’m not sure that it contains many additional informations to what is already described in specific articles already published. Therefore, if the goal of this paper ia to resume data regarding molecular traffic in the bladder inflammation I think it is very very important to achieve a more readable paper (better structured).

            The originality of this article is to compare the mechanisms of 3 forms of cystitis to propose common mechanisms. To our knowledge, no other review article has proposed this type of knowledge and proposed common mechanisms. In addition, this review article provides the reader with additional information for each type of cystitis, suggesting new avenues for investigation. The introduction has been modified to reinforce this message.

  1. I would try to speculate future therapeutic targets for cystitis basing on the data reported in this paper.

 A paragraph has been added on therapeutic targets. The therapeutic targets are illustrated in Figure 4 and discussed in the conclusion. However, the therapeutic targets are specific to a mechanism and do not allow reversing the pathology. It seems therefore necessary to act simultaneously on several mechanisms. Cellular therapy with mesenchymal stem cells seems to be the most promising and has proven its effectiveness in several forms of cystitis as presented in conclusion.

Reviewer 2 Report

  1. The use of the term 'cystitis' is somewhat misleading, as the authors discuss specific types of cystitis whereas the term is most commonly used for the commonest of the urinary tract infections. I propose that the authors revise the title and the text to reflect the specific use of the term in their manuscript
  2. The common pathogenetic pathways that the authors aim to present between the 3 types of cystitis are left to the readers to figure out , basically via a Table. I believe that an explanatory Discussion could/should have been added, so as to make their message clearer 
  3. The Conclusions actually present the hypotheis that the authors are trying to put forward. This hypothesis should go under a different section, possibly as part of a Discussion which is missing
  4. The Tables are really blurred and difficult to read. Please revise  

Author Response

We acknowledge the reviewer for the review and supportive guidance. We have cautiously considered the reviewer’s commentaries and information and revised our document consequently (highlighted in yellow).

  1. The use of the term 'cystitis' is somewhat misleading, as the authors discuss specific types of cystitis whereas the term is most commonly used for the commonest of the urinary tract infections. I propose that the authors revise the title and the text to reflect the specific use of the term in their manuscript

The title and text have been modified to specify the types of cystitis studied in this review article

  1.       The common pathogenetic pathways that the authors aim to present between the 3 types of cystitis are left to the readers to figure out , basically via a Table. I believe that an explanatory Discussion could/should have been added, so as to make their message clearer.

The paragraphs have been restructured to present first the common parts of each mechanism, then the specificity of each cystitis is presented.  At the end of the discussion, the common pathogenic pathways of the 3 types of cystitis and treatments are developed and associated with a figure to make their message clearer.

  1. The Conclusions actually present the hypothesis that the authors are trying to put forward. This hypothesis should go under a different section, possibly as part of a Discussion which is missing

A specific paragraph on hypothesis was placed in the discussion and illustrated by Figure 4.

  1. The Tables are really blurred and difficult to read. Please revise

 The tables have been reconstructed to improve readability and Table 2 has been separated into two tables (tables 2 and 3).

Round 2

Reviewer 1 Report

The manuscript has improved and I personally thank very much the Authors for having taken into considerations my suggestions. I try to further improve it all. 

I think that the purpose of this work is to simplify the understanding of the main pathophysiological mechanisms that determine the clinical features of cystitis. Therefore, I think that a greater simplicity, sequentiality and schematicity in the description of them could be essential in order not to confuse the reader.

At the beginning of the manuscript, 3 types of cystitis are represented: hemorrhagic, interstitial and chronic radiation. Subsequently, however, the description of the physiopathological mechanisms concerns only 2 of the 3. Paragraph 3.2 reports that generic and specific physiopathological mechanisms of cystitis will be examined. However, in the following paragraph this distinction is not clearly made. Therefore the reader does not find in the descriptive sequence what is preliminarily presented.

I would organize the work in a different way:

1) after a brief introduction describing the classification of cystitis, I would describe the clear objective of the study, which is to expose the pathophysiological mechanisms of abacterial cystitis;

2) I would briefly describe the etiological aspects of radiation cystitis and interstitial cystitis, the clinical features and currently used therapies;

3) then, I would move on to the description of their common molecular pathophysiological mechanisms and I would make a small separate paragraph to highlight the aspects for which the two forms differ;

4) on haemorrhagic cystitis, having common etiological and pathophysiological elements with the others, probably no specific treatment is needed but only a brief description.

Author Response

Dear Editors and reviewer,

We thank you again for these enriching comments and believe that after incorporating the reviewer's advice, the revised manuscript has been considerably strengthened and has gained in clarity. The reviewer's careful proofreading has uncovered some errors and confusions. We thank him for his strong involvement. A detailed point-by-point response to the reviewers' comments is attached. Changes to the original manuscript are indicated by underlined text.

The purpose of this work is to simplify the understanding of the main pathophysiological mechanisms that determine the clinical features of cystitis. Therefore, I think that :

  • A greater simplicity, sequentiality and schematicity in the description of them could be essential in order not to confuse the reader.

We tried to remove unnecessary sentences, add title to each paragraph, modify structure of paragraphs if necessary. Remove unnecessary sentences. Concerning the goal about concision, keep in mind that some paragraph such as “Structure and function of the bladder wall” are requested by reviewer 2 for a better understanding of mechanisms.

  • At the beginning of the manuscript, 3 types of cystitis are represented: hemorrhagic, interstitial and chronic radiation. Subsequently, however, the description of the physiopathological mechanisms concerns only 2 of the 3.

To eliminate this misunderstanding, we rewrite some sentences and add paragraph titles (in yellow)

  • Paragraph 3.2 reports that generic and specific physiopathological mechanisms of cystitis will be examined. However, in the following paragraph this distinction is not clearly made. Therefore, the reader does not find in the descriptive sequence what is preliminarily presented.

We apologize for this mistake. Paragraph 3.2 is in fact paragraph 4. For this paragraph, title and first sentence are changed.

I would organize the work in a different way:

  • After a brief introduction describing the classification of cystitis,

Introduction was shortened, and classification of cystitis was added.

  • I would describe the clear objective of the study, which is to expose the pathophysiological mechanisms of abacterial cystiti 

Objective was rewritten according to recommendation of reviewer

  • I would briefly describe the etiological aspects of radiation cystitis and interstitial cystitis, the clinical features and currently used therapies.

Etiological is mentioned for each pathology. In agreement with the other reviewer, therapies are described in conclusion.

  • Then, I would move on to the description of their common molecular pathophysiological mechanisms and I would make a small separate paragraph to highlight the aspects for which the two forms differ.

Specific paragraphs are added for common and other paragraph for specific mechanisms.

Reviewer 2 Report

Thank you for addressing all comments raised by this reviewer. I have no further comments.

Author Response

Article is accepted by reviewer 2